# Chondrosarcoma of the Proximal Humerus: Does the Margin Affect Survival?

**DOI:** 10.3390/cancers15082337

**Published:** 2023-04-17

**Authors:** Gilber Kask, Minna K. Laitinen, Michael C. Parry, Jose I. Albergo, Jonathan D. Stevenson, German Farfalli, Luis Aponte-Tinao, Robert Grimer, Vaiyapuri Sumathi, Lee M. Jeys

**Affiliations:** 1Department of Orthopaedics and Traumatology, Helsinki University Hospital, University of Helsinki, 00100 Helsinki, Finland; 2Royal Orthopaedic Hospital, Birmingham and Aston University Medical School, Aston University, Birmingham B4 7ET, UK; 3Department of Orthopaedics, Hospital de Italiano, Buenos Aires C1199, Argentina; 4Unit of Oncology, Royal Orthopaedic Hospital, Birmingham B31 2AP, UK; 5Royal Orthopaedic Hospital, Faculty of Health Sciences, Aston University, Birmingham B4 7ET, UK

**Keywords:** chondrosarcoma, malignant tumours, bone tumours, proximal, humerus, survival

## Abstract

**Simple Summary:**

The aim of this study was to investigate factors that affect the local and systemic prognoses for conventional, central CSs (those that arise from within the medulla of the bone) of the proximal humerus. Our results show that proximal humeral grade 1 CSs behave as a more benign tumour, having no cases of LR nor death due to disease. Grade 2 CSs of the proximal humerus act as a low-grade tumour, being locally aggressive with higher rates of LR than grade 1 CSs but still having a low incidence of mortality and high rates of DSS. The LR does not affect the DSS; therefore, the surgical management in proximal humeral grade 2 CSs should have a greater focus on function but still aim for a margin free resection.

**Abstract:**

Chondrosarcoma (CS) is the second most common primary malignant bone tumour and, in the absence of reliable chemotherapy and radiotherapy, is effectively a surgical disease. Overall disease specific survival (DSS) is affected by tumour grade, whilst resection margin contributes to local recurrence free survival (LRFS). The aim of this study was to investigate factors that affect the local and systemic prognoses for conventional central CSs arising from the proximal humerus. A multi-centre, retrospective study from three international collaborative sarcoma centres identified 110 patients between 1995 and 2020 undergoing treatment for a conventional central CS of the proximal humerus; 58 patients (53%) had a grade 1 tumour, 36 (33%) had a grade 2 tumour, and 16 patients (13%) had a grade 3 CS. The mean age of patients was 50 years (range 10–85). The incidence of local recurrence (LR) was 9/110 (8.2%), and the disease specific mortality was 6/110 (5.5%). The grade was a statistically significant factor for LRFS (*p* < 0.001). None of the grade 1 tumours developed LR. The DSS was affected by the grade (*p* < 0.001) but not by the LR (*p* = 0.4). Only one patient with a grade 2 tumour died from the disease. The proximal humeral grade 1 CS behaved as a benign tumour, having no cases of LR nor death due to disease. Grade 2 CSs of the proximal humerus behaved in a more indolent way when compared with comparable grade tumours elsewhere in the appendicular skeleton, being locally aggressive with a higher LR rate than grade 1 CSs but still having very low mortality and a high rate of DSS. The LR in grade 2 CSs did not affect the DSS; therefore, surgical management in proximal humeral grade 2 CSs should have a greater emphasis on preserving function whilst maintaining an adequate margin for resection. The proximal humeral grade 3 CS was, as elsewhere in the skeleton, an aggressive, high-grade tumour. Therefore, surgical management should include en bloc resection with clear margins to avoid LR.

## 1. Introduction

Chondrosarcoma (CS) is the second most common primary malignant tumour of bone, accounting for approximately 20% of all bone sarcomas [1]. CS is a surgical disease due to its resistance to chemotherapy and radiotherapy [2,3]. The surgical management of CS must take into consideration the grade of the tumour, as well as the stage at the diagnosis, all of which are known to influence the overall disease specific survival (DSS) [4]. Consideration must be given to the projected margin that can be achieved at surgical resection as this too has been shown to significantly affect the incidence of local recurrence (LR) and the overall survival [5].

CSs are graded as low (grade 1) and high (grades 2 and 3). The grade 1 CS has more recently been re-classified as an atypical cartilaginous tumour (ACT) [6], an intermediate locally aggressive tumour, which more accurately reflects its unique tumour characteristics. However, since the reclassification, the incidence of ACTs has significantly increased [7].

The anatomical location and grade of the tumour are known to affect the incidence of LR and overall disease specific survival (DSS). High-grade CSs of the axial skeleton have a worse prognosis when compared with CSs of the extremities, regardless of the grade [2,8,9]. In the extremities, grade 1 CSs/ACTs rarely metastasize, conferring an excellent prognosis for survival [10,11,12,13,14,15,16]; they also have a low rate of LR [11,12,15], such that they can be safely treated with intralesional curettage [8,12,14,17]. In the proximal humerus, the second most common appendicular site for CSs after the proximal femur [18], intralesional curettage to preserve the integrity of the rotator cuff, thus preserving function, is a particularly attractive surgical option [19]. Appendicular grade 2 and 3 CSs are classified as high-grade tumours that have a propensity for metastases and rapid growth [8,17]. In such cases, intralesional surgery confers a high incidence of LR, which has been shown to negatively affect overall survival [9]. However, grade 2 CSs of the hands and feet behave in a more benign, indolent way when compared with comparable grade tumours in other locations [20]. Within the appendicular skeleton, variations in the biological behaviour of high-grade CSs can be seen. For example, DSS and LR free survival (LRFS) have been shown to be superior for upper extremity CSs compared with those arising from the pelvis or lower extremities [21]. Therefore, to tailor the surgical management of CS to an individual patient, surgeons must take into consideration all these factors when deciding how best to manage a specific tumour.

The purpose of this study was to investigate the factors that may affect the local and systemic prognoses for central conventional CSs arising from the proximal humerus.

## 2. Materials and Methods

Following institutional ethical review board approval, patients who were diagnosed and surgically treated for a CS in the proximal humerus between January 1995 and January 2020 at three large tertiary referral sarcoma centres, in three different countries (the Royal Orthopaedic Hospital, Birmingham, UK; Hospital Italiano, Buenos Aires, Argentina; and Helsinki University Hospital, Helsinki, Finland), were identified from retrospective institutional databases. All patients were diagnosed and treated at the referral hospital. Those who were primarily treated elsewhere and referred for the management of a recurrent tumour were excluded. A minimum of two years of follow-up was required for survivors. All patients had continuous follow-up until the point of the last clinical assessment or the time of death. Details of the clinical data and oncological outcomes, including the LRFS and DSS, were collected. The primary surgery was defined by the method that concluded the first-line treatment. The resection specimens were examined by specialist bone sarcoma pathologists, for grade and margin status in each centre, defined by internationally agreed-upon standards and described according to the WHO classification. The margin was quantified by a specialist bone sarcoma pathologist and classified according to the system described by Enneking [22]. The reported smallest margin was in the soft tissue. Histologic grades were determined based on cellularity, nuclear size, and the presence of an abundant hyaline cartilage matrix or mucomyxoid matrix and mitoses. The highest grade seen on the histology was the grade recorded, even when this higher grade comprised a small number of cells. The histological diagnosis and treatment plan according to grade and radiology were made by a multidisciplinary team in a consensus meeting in each hospital. A complete dataset was available for all patients included in the final analysis (Figure 1).

The primary outcome measure was LRFS, and the secondary outcome measures were predictors of LR and DSS.

Continuous variables were reported as mean and range. LRFS rates, including 95% CI, were assessed using the Kaplan–Meier method. Survival rates were calculated from the date of the surgery to the most recent follow-up, confirmation of LR, or death. LR was defined as tumour relapse according to radiographic evidence, later confirmed histologically or by the presence of a growing tumour mass radiologically. The criteria for the definition of LR were the same in all centres. Between-group comparisons were performed using the log-rank test. The Cox regression model was used to identify independent factors affecting the LRFS. The differences in the proportions were assessed using Fisher’s exact test. The age was normally distributed and tested by the Shapiro–Wilk test.

The subdistribution hazard ratio (SHR) of the role of different factors on survival was calculated using competing risk analysis. Competing risk analysis aims to correctly estimate the marginal probability of an event in the presence of competing events. Death due to another reason was considered as a competing event in the analysis of the role of different factors on the DSS. Synchronous metastases (metastases that developed before LR, at the time of LR, or within 90 days after LR) and death due to another reason were considered as competing events in the analysis of the role of LR on DSS. All other statistical analyses were completed using SPSS Statistics 24.0 (IBM, New York, NY, USA), but competing risk analysis was performed using STATA 17 (Stata, College Station, TX, USA).

## 3. Results

The final study population comprised 110 patients with proximal humeral CSs, of which 58 (53%) had grade 1 tumours, 36 (33%) had grade 2, and 16 (13%) had grade 3 CSs. The mean age of the study population was 50 years (10–85 years), and the median follow-up time was 84 months (6–360 months). Seven patients died before two years of follow-up, four due to sarcoma and three due to other causes. The patients’ characteristics are presented in Table 1.

### 3.1. Predictors of LRFS and LR

The overall incidence of LR was 8.2% (9/110 patients). The LRFS for all tumours was 97.2% (95% CI: 94–100) at 1 year, 94.1% (95% CI: 89–99) at 3 years, 92.9% (95% CI: 88–98) at 5 years, and 88.9% (82–96) at 10 years. In grade 1 CSs, the LRFS was 100% at 1, 3, 5, and 10 years. In grade 2 CSs, the LRFS was 100% at 1 year, 93.5% (95% CI: 85–100) at 3 and 5 years, and 89.3% (95% CI: 78–100) at 10 years. In grade 3 CSs, the LRFS was 80.8% (95% CI: 61–100) at 1 year, 74.0% (95% CI: 52–96) at 3 years, and 61.7% (95% CI: 33–90) at 5 and 10 years.

In each case where LR occurred, the grade of the LR was the same as that of the initial, primary tumour. LR in grade 2 CSs appeared at 17, 36, 64, and 94 months, over periods of follow-up of 268, 107, 120, and 237 months, respectively.

The grade was a statistically significant factor for LRFS (*p* < 0.001) (Figure 2). None of the other factors studied were statistically significant after univariate survival analysis, neither when all grades were combined nor after stratifying according to grade.

The margin was not a statistically significant predictor for LRFS, neither when all grades were combined (*p* = 0.8) nor when stratified according to grade (grade 1, *p* = 1; grade 2, *p* = 0.4; or grade 3, *p* = 0.1). The role of LR is summarized in Table 2.

### 3.2. Disease Specific Survival

The overall disease specific death rate was 5.5% (six of 110 patients). The DSS was 100% at 1 year, 95.0% (95% CI: 91–99) at 3 and 5 years, and 93.4% (95% CI: 87–99) at 10 years. In grade 1 CSs, the DSS was 100% at 1, 3, 5, and 10 years. In grade 2 CSs, the DSS was 100% at 1 year and 97.1% (95% CI: 0.91–100) at 3, 5, and 10 years. In grade 3 CSs, the DSS was 100% at 1 year, 71.4% (95% CI: 48–95) at 3 and 5 years, and 57.1% (95% CI: 26–89) at 10 years. In the competing risk analysis, the tumour grade was a statistically significant factor affecting DSS (*p* = 0.002, SHR 16.2; 95% CI: 2.7–95.5) (Figure 3). Other significant factors were the tumour size (*p* = 0.004, SHR 1.13; 95% CI: 1.04–1.23), the presence of metastases at the diagnosis (*p* < 0.001, SHR 2.14 × 10^15^, 95% CI: 8.61 × 10^13^–5.33 × 10^6^), and LR (*p* < 0.001, SHR 12.8; 95% CI: 2.9–57.2). However, when stratified by grade, LR was not a statistically significant factor for DSS. No patients with a grade 2 CS who developed LR died of the disease. One patient who died as a result of their grade 2 CS (one out of 36) had metastases at the time of the diagnosis and prior to surgical treatment and had no evidence of LR prior to death due to metastatic disease (Figure 4).

The marginal status was not a significant factor for survival (*p* = 0.553, HR 1.4; 95% CI: 0.5–3.6) (Figure 5).

## 4. Discussion

Conventional chondrosarcoma, whilst regarded as a single disease entity, constitutes a broad spectrum of tumour variants that behave in different ways depending on different clinical features. Of these variables, the most important predictor influencing survival is the grade [16]. However, it is becoming more apparent that the location of the tumour also impacts survival. This is perhaps most evident in the pelvis, where CS is known to have a worse prognosis [9] when compared with acral or appendicular locations [20]. Our results, when compared with evidence from the literature, demonstrate that proximal humeral CSs behave in a more indolent, less aggressive manner than comparable tumours in other extremity locations [8]. Awareness of this clinical variability will hopefully allow a more tailored approach to the management of grade 1 and 2 tumours with improvements in function as a result of a more conservative approach, where appropriate.

Within the study population, we have demonstrated that higher tumour grade and the presence of metastasis are independent factors predicative of worse DSS, consistent with the available literature. The tumour grade affects the rate of LR and thereby LRFS, but our results suggest we have shown the presence of LR is not associated with a worse prognosis nor a decrease in DSS, either when grade is excluded or when stratified according to grade. Whilst much evidence exists within the literature of the behaviour of chondrosarcomas as a whole disease entity, there is little evidence relating to chondrosarcomas at specific anatomical sites, the assumption being, certainly historically, that chondrosarcomas of comparable grades within the appendicular skeleton, at least, all behaved in a comparable way. We have shown, however, that chondrosarcomas of the proximal humerus appear to display a more indolent, less aggressive behaviour. We have shown, in accordance with Mourikis et al., an incidence of metastatic disease of 5.5% and overall DSS of 96.8% after 1 year, 88.0% at 3 and 5 years, and 86.4% at 10 years [13]. This is in stark contrast to the DSS seen for chondrosarcomas arising within the pelvis, which is estimated as 95% for grade 1 tumours, 70% for grade 2, and 50% for grade 3 tumours at 5 years and 95% for grade 1, 65% for grade 2, and 35% for grade 3 at 10 years [9].

Our results also demonstrate that in grade 1 CSs, none of the patients developed metastases or died of the disease, which is in accordance with the previous literature [11,16,23,24]. Consequently, intralesional curettage is the widely accepted treatment in proximal humeral grade 1 CSs [24,25]. According to the literature, the rates of LR after curettage in grade 1 CSs vary between 9% [25,26] and 44% [9]. However, we did not identify any incidence of LR in grade 1 tumours after curettage, indicating a more benign local behaviour. We have previously shown the effect of accurate grading on the incidence of LR and the overall survival, in particular, the challenge of differentiating between true low-grade chondrosarcomas and those with focal areas of higher grade. We have shown that the disease behaviour is based entirely on the highest grade seen in the final resection or curettage specimen. This concept is relatively new, and perhaps this finding demonstrates a retrospective application of this principle [16]. In years gone by, it may be that pathologists gave an overall assessment of the grade, and where there were only a few areas approaching higher grade, the predominant background grade was quoted. We now know that even when seen in only a small area, the tumour will behave at the highest grade seen. Therefore, perhaps what we see here are the results for purely low-grade chondrosarcomas, without any areas of higher grade. The effect of the location may be more difficult to answer in so much as why the rate of LR is lower for purely low-grade chondrosarcomas in the proximal humerus when compared with purely low-grade chondorosarcomas elsewhere. The challenge, again, is on predicting from pre-operative imaging the grade of the tumour, which can be extremely challenging.

The outcomes for patients with low-grade chondrosarcoma may be an effect of the treatment strategy. In 54% of cases of low-grade tumours, a resection was undertaken at the outset, with a wide margin achieved in 36%. This may, in part, explain the low rate of LR but of course does not explain the low rate of LR seen in low-grade tumours treated with marginal or intralesional surgery where the incidence of LR was equally low. An explanation, therefore, must reflect a difference in tumour behaviour between grades and also an effect of the location within the proximal humerus.

Our results show an overall LR rate of 8%, which is in line with the previous study relating to CSs of the proximal humeral (13%) [13]. However, in contrast to previous reports, we were unable to correlate LR with an effect of DSS in grade 2 CSs [5]. Only one patient with a grade 2 CS developed metastases and eventually died of their disease, however, without LR. The low number of patients may have influenced this result, but we were unable to show any difference in the rate of LR between different margins, neither by combining all grades together nor after stratification by grade. We were able to demonstrate, however, that improved margins reduced the incidence of LR. Forequarter amputation and disarticulation, ablative options that were associated with wider margins, were associated with a reduced incidence of LR but did not translate to an improved overall survival. This, of course, should come as no surprise. The largest tumours, displaying the most aggressive radiological features at presentation, are invariably those with the highest grades. It is exceptionally unlikely that a low-grade CS will present with such a burden of disease at the proximal humerus that the treating surgeon would consider that the only option to achieve an adequate margin would be an amputation. Therefore, again, this highlights the dissociation between the grade, LR, and overall survival.

It appears that in the proximal humerus, the margin status achieved at resection for grade 2 CSs is far less defined than for other body sites. We demonstrated that even marginal resections of a grade 2 CS were associated with a comparatively low incidence of LR. Conversely, radical or wider margins as were seen for grade 2 CSs treated by forequarter amputation or shoulder disarticulation did not translate into an improved overall survival.Therefore there remains a question about how far we should go to achieve a wide soft tissue margin. If a clear margin around nerve is achievable, the inference from the authors is that the nerve could be spared.

We have previously demonstrated a poor correlation between the highest grade seen on pre-operative biopsy and final resection histology, which at best, is estimated to be 50%, bringing into question the role of the pre-operative biopsy in guiding the planned resection strategy [16]. Our results for CSs of the proximal humerus, perhaps, make this decision-making strategy even more challenging. In accordance with Ma et al., we have shown that grade 2 CSs are locally more aggressive, with a higher rate of LR than grade 1 CSs, but the DSS for grade 2 CSs of the proximal humerus appears similar to that of grade 1 CSs, which contrasts to previous studies, where grade 2 CSs were considered as a high-grade tumour [6]. The reason for this less aggressive behaviour of grade 2 proximal humeral CSs remains unclear but is in keeping with the variable biological behaviour of CSs seen in different anatomical locations. The histological grading of these tumours, which underpins the overall results, may of course be subject to interobserver variability, which may, in part, account for some of the differences seen. However, the findings of this study, from three separate sarcoma centres, appear consistent across these centres. That is to say, whilst the numbers for any one of the three centres taken in isolation are too small to produce any meaningful results, the findings are consistent and, when combined, demonstrate significance. In our study, the histological grade was assessed by experienced bone tumour pathologists on the resection specimen, and if the histology showed a mixed grade, e.g., grade 1–2, the highest grade was taken as the definitive grade, as per previous studies. A multidisciplinary approach integrating the clinical and radiological features in combination with a histological assessment was used to ensure reliable grading of the tumours. Therefore, it is our opinion that the histologic grading was reliably distinguished. The less aggressive behaviour of grade 2 CSs resembles more the behaviour of CSs of the hand, where it has been classified as a ‘non-metastasizing’ tumour by some authors [27].

Advancing patient age has been reported as an independent risk factor for worse survival and disease recurrence in pelvis CS patients [28]. In our study, after stratification by grade, age was identified as a significant risk factor for a worse prognosis in grade 3 CSs of the proximal humerus only. The presence of metastases at the diagnosis and tumour grade remained statistically significant independent prognostic factors for poor DSS. The incidence of metastases and LR were 31% in grade 3 CSs; furthermore, LR did not have a statistically significant effect on DSS, reflecting the aggressive nature of grade 3 CSs, which manifests as a high incidence of systemic relapse and high rates of mortality.

This study is not without limitations, which include those inherent to its retrospective design. Even though this is the largest study on CSs of the proximal humerus, the small number of patients included in each group may have influenced the results as statistical significance is difficult to achieve. The inclusion criteria were limited to patients in whom radiological and pathological data were complete. This will have influenced the time frame for inclusion and follow-up for the study. However, the excluded patients were mostly treated prior to the accrual years, and since 2020, all consecutive patients in each centre had comprehensive data and were included in this study. Since the literature about CS in proximal humerus is very limited, and we had to compare our results to the results from all appendicular locations. Therefore, in the future, our results require validation with a larger dataset. Moreover, this was the largest study on conventional central proximal humerus CSs with accurate patient data. Since chondrosarcoma is commonly seen in more senior patients, death due to other diseases frequently occurs prior to death from CS. Therefore, our statistical method of using survival data calculations with competing risk analysis gives a more accurate reflection of the effect of CS on overall survival.

## 5. Conclusions

In conclusion, CSs of the proximal humerus seemed to behave in a less aggressive manner when compared with CSs in other extremity locations, when compared with the literature. Grade 1 CSs behaved in an indolent fashion, having low rates of LR with no patients going on to succumb from their disease. Grade 2 CSs behaved more like a low-grade tumour when arising from the proximal humerus being locally aggressive with a higher rate of LR but still having a low incidence of disease specific mortality, which was not affected by the development of LR. This behavioural anomaly displayed by grade 2 CSs of the proximal humerus must be taken into consideration when planning surgical management, with perhaps a greater focus being placed on function than one may have considered when managing grade 2 CSs at other anatomical locations. Grade 3 CSs remained aggressive, high-grade tumours even in the proximal humerus, where management remained the complete removal of the tumour with clear margins to achieve an acceptable incidence of LR. Even with such an approach, the DSS for high-grade CSs remained low.

## Figures and Tables

**Figure 1 cancers-15-02337-f001:**
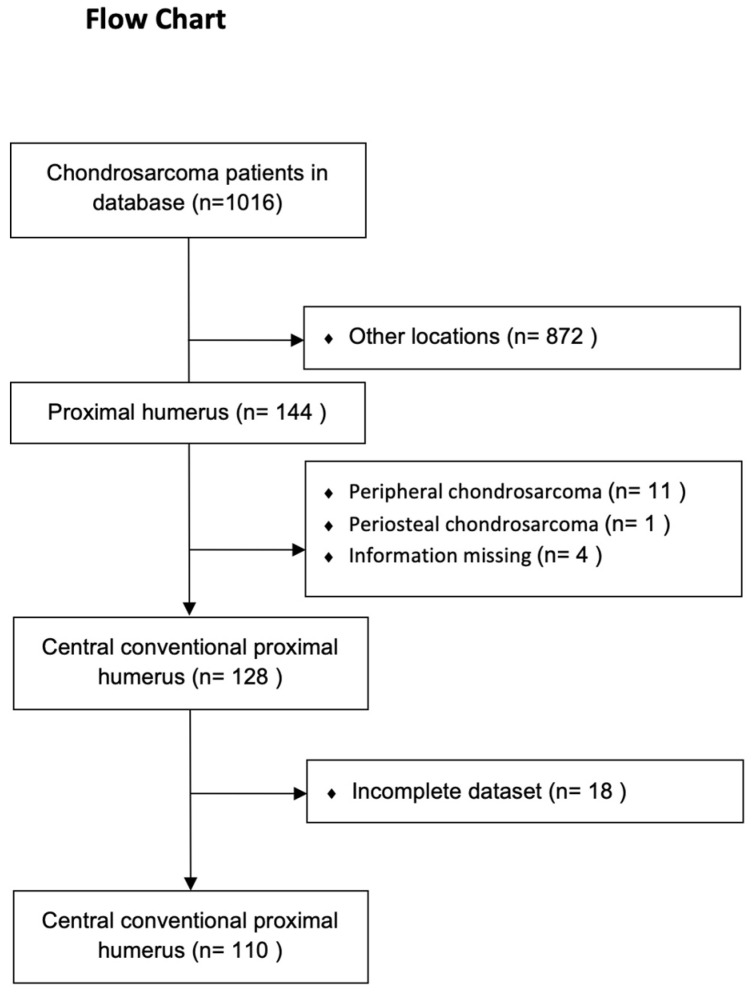
Flowchart of identification of patients with conventional central chondrosarcomas of the proximal humerus.

**Figure 2 cancers-15-02337-f002:**
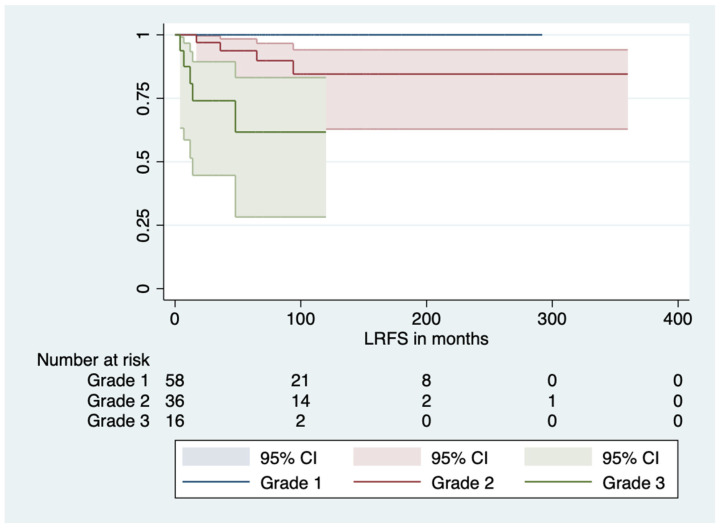
The role of grade in local recurrence free survival.

**Figure 3 cancers-15-02337-f003:**
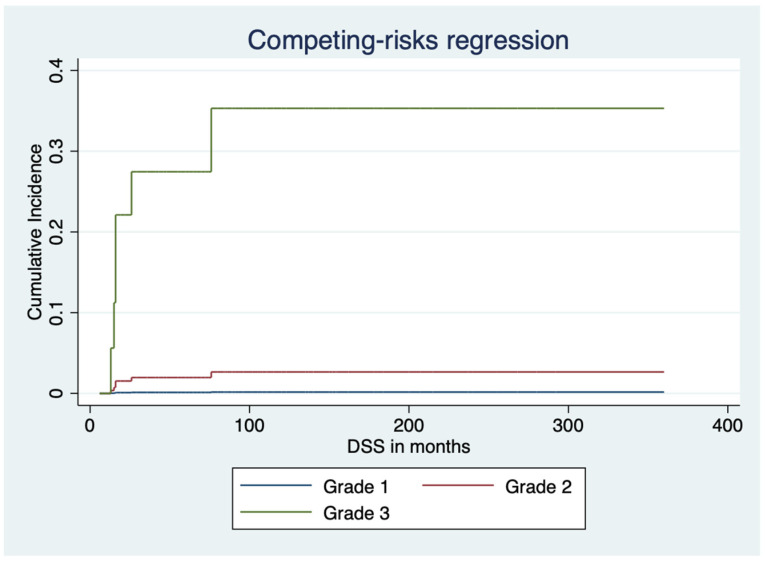
The role of grade in disease specific failure studied with a competing risk model.

**Figure 4 cancers-15-02337-f004:**
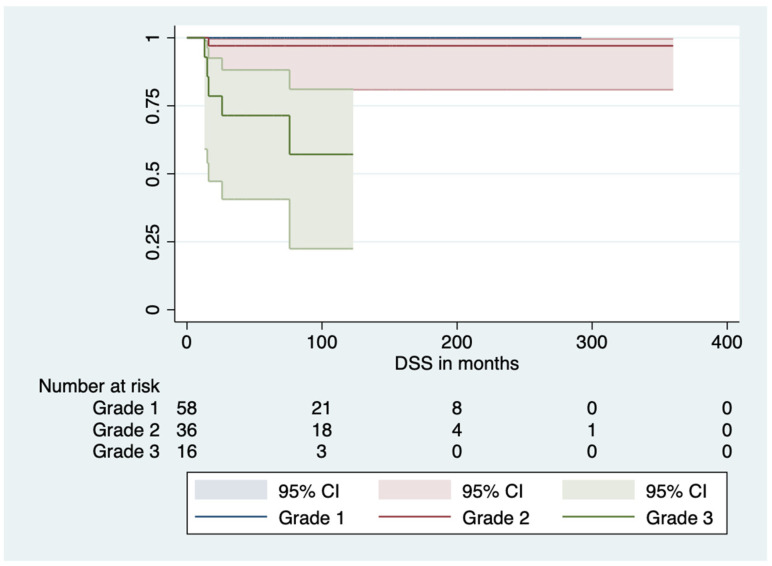
Only one patient with a grade 2 tumour died of the disease.

**Figure 5 cancers-15-02337-f005:**
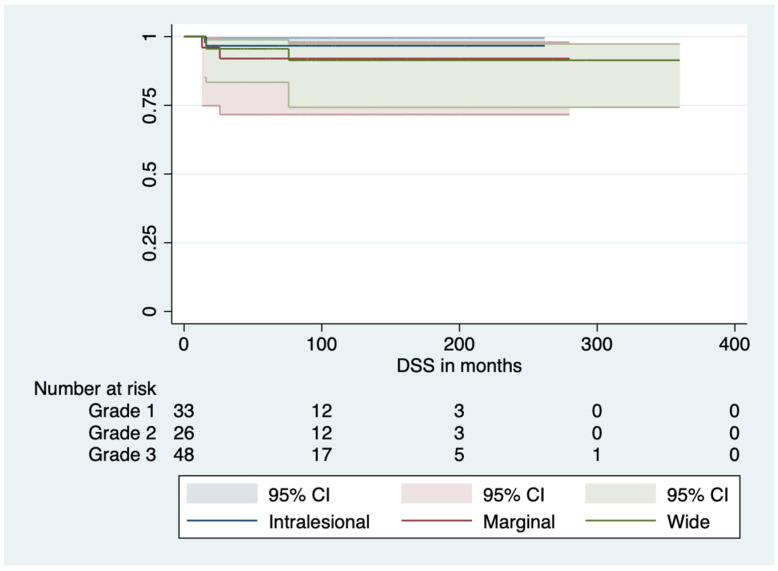
Margin was not significant in disease specific survival.

**Table 1 cancers-15-02337-t001:** Characteristics of 110 proximal humerus central conventional chondrosarcoma cases (values are presented as number of cases).

Characteristics	Total	Grade 1	Grade 2	Grade 3
Eligible cases	110	58 (53%)	36 (33%)	16 (14%)
Sex				
Male	47 (43%)	21 (36%)	16 (44%)	10 (63%)
Female	63 (57%)	37 (64%)	20 (56%)	6 (37%)
Median age at surgery (range)	50 (10–85)	50 (19–73)	48 (10–85)	64 (40–85)
Median tumour size, cm (range)	9.0	7.0	8.5	14.0
	(1.3–28)	(1.3–25)	(4.0–26)	(8.5–28)
Surgery				
Curettage	27 (24%)	26 (45%)	1 (3%)	-
Resection	77 (70%)	31 (54%)	34 (94%)	12 (75%)
Amputation	4 (4%)	1 (2%)	1 (3%)	2 (13%)
Forequarter	2 (2%)	-	-	2 (13%)
Margin				
Wide	51 (46%)	21 (36%)	21 (58%)	9 (56%)
Marginal	26 (24%)	10 (17%)	10 (28%)	6 (37%)
Intralesional	33 (30%)	27 (47%)	5 (14%)	1 (6%)
Extraosseous component	26 (61%)	-	15 (48%)	11 (69%)
Data missing	3	0	3	0
Median follow-up, months (range)	84 (6–360)	85 (6–292)	98 (7–360)	40 (11–123)
Pathologic fracture				
No	71 (83%)	38 (84%)	26 (87%)	7 (64%)
Yes	15 (17%)	7 (16%)	4 (13%)	4 (36%)
Data missing	24	13	6	5
Metastasis *				
No	103 (94%)	58 (100%)	34 (94%)	11 (69%)
Yes	7 (6%)	-	2 (6%)	5 (31%)
Time (mean) to metastasis in months (range)	10.0 (0–27)	-	10.5 (10–11)	5.0 (0–27)
Local recurrence				
No	101 (92%)	58 (100%)	32 (89%)	11 (69%)
Yes	9 (8%)	-	4 (11%)	5 (31%)
Median time to LR in months (range)	17 (4–94)	-	50 (17–94)	12 (4–48)
Syndromes				
None	104 (95%)	57 (98%)	33 (94%)	14 (87%)
Ollier’s	5 (5%)	1 (2%)	2 (6%)	2 (13%)
Dead from disease				
No	104 (95%)	58 (100%)	35 (97%)	11 (69%)
Yes	6 (5%)	-	1 (3%)	5 (31%)

* Metastasis at diagnosis or later.

**Table 2 cancers-15-02337-t002:** Nine local recurrences of 110 conventional central proximal humerus chondrosarcoma cases (values are presented in number of cases).

Characteristics	Total	No LR	LR	*p* Value
Eligible cases	110	101 (92%)	9 (8%)	
Sex				0.4 #
Male	47 (43%)	42 (42%)	5 (56%)	
Female	63 (57%)	59 (58%)	4 (44%)	
Age at surgery, median (range)	56 (10–85)	56 (10–85)	57 (31–85)	0.7 ±
Median tumour size, cm (range)	10.5 (4.0–27.5)	9.0 (4.0–27.5)	12.5 (7.0–19.0)	0.8 ±
Surgery				0.2 #
Curettage	27 (25%)	27 (27%)	0 (0%)	
Resection	77 (70%)	68 (67%)	9 (100%)	
Amputation	4 (4%)	4 (4%)	0 (0%)	
FQ	2 (4%)	2 (5%)	0 (0%)	
Margin				0.1 #
Wide	51 (46%)	48 (47%)	3 (33%)	
Marginal	26 (24%)	21 (21%)	5 (56%)	
Intralesional	33 (30%)	32 (32%)	1 (11%)	
Tumour grade				**<0.001 #**
G1	58 (53%)	58 (57%)	-	
G2	36 (33%)	32 (32%)	4 (44%)	
G3	16 (14%)	11 (11%)	5 (56%)	
Extraosseal component				**<0.001 #**
No	79 (45%)	78 (51%)	1 (13%)	
Yes	26 (55%)	19 (49%)	7 (88%)	
Data missing	5	4	1	
Median follow-up, months (range)	84 (6–360)	84 (6–360)	107 (12–268)	0.7±
Pathologic fracture				0.4#
No	71 (83%)	66 (84%)	5 (71%)	
Yes	15 (17%)	13 (16%)	2 (29%)	
Missing	24	6	0	
Metastasis *				
No	103 (94%)	97 (96%)	6 (67%)	**0.01** #
Yes	7 (6%)	4 (4%)	3 (33%)	
Syndromes				0.4 #
None	105 (95%)	97 (96%)	8 (89%)	
Ollier’s	5 (5%)	4 (4%)	1 (11%)	
Dead from disease				**0.002** #
No	104 (95%)	98 (97%)	6 (67%)	
Yes	6 (5%)	3 (3%)	3 (33%)	

Analysed using Pearson’s chi-square (#) and ANOVA (±). * Metastasis at diagnosis or later.

## Data Availability

Deidentified patient data are available from the authors.

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
