# Peer review of "Chondrosarcoma of the Proximal Humerus: Does the Margin Affect Survival?"

_cancers, 2023, doi:10.3390/cancers15082337_

Round 1

Reviewer 1 Report

This is an international multi-centre, retrospective study from three large tertiary referral sarcoma centres, (the Royal Orthopaedic Hospital, Birmigham, UK, Hospital Italiano, Buenos Aires, Argentina, and Helsinki University Hospital, Helsinki, Finland).

The aim of this study was to investigate factors which affect the local and systemic prognosis for conventional central CS arising from the proximal humerus. The study identified 110 patients from retrospectively reviewed, prospectively validated institutional databases, between 1995 and 2020.

Main conclusion in the article:

Grade 1 CS behaves in an indolent fashion, having low LR rate and none of the patients died of their disease

·       Grade 2 CS behave more like a low-grade tumour in the proximal humeral location, being locally aggressive with a slightly higher LR rate, but still having very low disease specific mortality rate and LR does not impact DSS

·      Grade 3 CS remains an aggressive high-grade tumor even in the proximal humerus, where with en-bloc local resection, DSS rate is low.

 Comments:

The authors should be acknowledged for obtaining a new understanding of this rare and complex disease. It is to compliment the focus on the humeral location, both for the lack of literature and for the clinical relevance. The paper is well written and easy understandable. The methods for data collection are well reported, and the statistical analysis with log-rank test, Cox regression, Fisher’s exact test and SHR are appropriate.

As the data is retrieved retrospectively, the study has its limitation, which has been sufficiently addressed in the discussion. The fact that the study is multi-centre is appreciated, both because of the rare nature of the disease and for inter-observer reliability. Methods and results are in accordance with good scientific practice and are well described and I only have few comments to improve the paper.

·      In line 25 the authors report “central osteosarcoma”. Please add a definition of central, acral and appendicular osteosarcoma. 

·        The title reflects only to a certain degree the article and is overselling the message of the paper a bit - if the authors do not state the previous data about amputation due to CS2 - and was amputation for CS2 in the humerus more common before 1995? Line 201, the authors state “Also, forequarter amputation and disarticulation resulted in wider margins and reduced LR but without improved survival. Therefore, achieving wider margins with mutilating disarticulation or forequarter amputation did not improve survival in proximal humeral grade 2 CS, since marginal margins achieved a low LR rate”. Is this assumption based on your study? You only have one patient treated with amputation in your CS2 group, and no forequarter amputation.

Please rewrite and down scale the research message in the title.

Line 101: please specify if the pathological evaluation is done in every center accordingly to WHO classification.

Line 110: how is LR confirmed? Histologically or with images?

Line 190, the authors state:” According to the literature, the rate of LR after curettage in grade 1 CS vary between 9% [24, 25] to 44% [9]. However, very surprisingly, we did not discover any LR in the grade 1 group after curettage, indicating benign local behavior.” Will you please discuss the reason for this found in your opinion? Could it be because you excluded patients primarily treated elsewhere? Could you maybe report how many patients you have excluded due to previous treatment and referred to your center due to recurrence?

Is it possible that the pathological evaluation of the samples and the classification to grade 2 even if the sample is mostly grad 1 + a small number of grad 2 cells has influenced the different results compared to the previous literature? Please elaborate further.

Minor details:

Line 41: p=0,000 should be reported as p=0 and p=0,397 as p=0,4

Line 125:” The mean age of the study population was 49 years (10-85 years) and the mean follow-up time was 92 months (4-360 months).You report a mean age of 48 in the abstract (line 38).

Table 2 description states, “8 local recurrences” and in the table are reported 9.

Author Response

Reviewer 1

This is an international multi-centre, retrospective study from three large tertiary referral sarcoma centres, (the Royal Orthopaedic Hospital, Birmigham, UK, Hospital Italiano, Buenos Aires, Argentina, and Helsinki University Hospital, Helsinki, Finland).

The aim of this study was to investigate factors which affect the local and systemic prognosis for conventional central CS arising from the proximal humerus. The study identified 110 patients from retrospectively reviewed, prospectively validated institutional databases, between 1995 and 2020.

Main conclusion in the article:

Grade 1 CS behaves in an indolent fashion, having low LR rate and none of the patients died of their disease

  • Grade 2 CS behave more like a low-grade tumour in the proximal humeral location, being locally aggressive with a slightly higher LR rate, but still having very low disease specific mortality rate and LR does not impact DSS

  • Grade 3 CS remains an aggressive high-grade tumor even in the proximal humerus, where with en-bloc local resection, DSS rate is low.

 Comments:

The authors should be acknowledged for obtaining a new understanding of this rare and complex disease. It is to compliment the focus on the humeral location, both for the lack of literature and for the clinical relevance. The paper is well written and easy understandable. The methods for data collection are well reported, and the statistical analysis with log-rank test, Cox regression, Fisher’s exact test and SHR are appropriate.

As the data is retrieved retrospectively, the study has its limitation, which has been sufficiently addressed in the discussion. The fact that the study is multi-centre is appreciated, both because of the rare nature of the disease and for inter-observer reliability. Methods and results are in accordance with good scientific practice and are well described and I only have few comments to improve the paper.

  • In line 25 the authors report “central osteosarcoma”. Please add a definition of central, acral and appendicular osteosarcoma.

In this sentence central chondrosarcoma refers to the intramedullary location of the tumour in contrast to peripheral chondrosarcoma which is secondary to a preexisting osteochondroma. The definition is added.

  • The title reflects only to a certain degree the article and is overselling the message of the paper a bit - if the authors do not state the previous data about amputation due to CS2 - and was amputation for CS2 in the humerus more common before 1995? Line 201, the authors state “Also, forequarter amputation and disarticulation resulted in wider margins and reduced LR but without improved survival. Therefore, achieving wider margins with mutilating disarticulation or forequarter amputation did not improve survival in proximal humeral grade 2 CS, since marginal margins achieved a low LR rate”. Is this assumption based on your study? You only have one patient treated with amputation in your CS2 group, and no forequarter amputation.

Please rewrite and down scale the research message in the title.

We thank the reviewer for this comment. We do agree that the title was overselling this article and we have changed it.

Line 101: please specify if the pathological evaluation is done in every center accordingly to WHO classification.

Thank you for this important comment. Indeed the pathologic evaluation and grading is cone is every centre according to WHO classification. This information has been added to the manuscript.

Line 110: how is LR confirmed? Histologically or with images?

LR was defined radiographically and later confirmed histologically or by serial imaging confirming a growing tumour mass.

Line 190, the authors state:” According to the literature, the rate of LR after curettage in grade 1 CS vary between 9% [24, 25] to 44% [9]. However, very surprisingly, we did not discover any LR in the grade 1 group after curettage, indicating benign local behavior.” Will you please discuss the reason for this found in your opinion? Could it be because you excluded patients primarily treated elsewhere? Could you maybe report how many patients you have excluded due to previous treatment and referred to your center due to recurrence?

Is it possible that the pathological evaluation of the samples and the classification to grade 2 even if the sample is mostly grad 1 + a small number of grad 2 cells has influenced the different results compared to the previous literature? Please elaborate further.

This is quite a difficult question to answer and we can, at best speculate on the reasoning. We have added a short paragraph to give a possible explanation but we acknowledge that this finding is difficult to answer beyond speculation.

Minor details:

Line 41: p=0,000 should be reported as p=0 and p=0,397 as p=0,4

The numbers have been changed accordingly

Line 125:” The mean age of the study population was 49 years (10-85 years) and the mean follow-up time was 92 months (4-360 months).” You report a mean age of 48 in the abstract (line 38).

Thank you for highlighting this inaccuracy. The numbers have been changed.

Table 2 description states, “8 local recurrences” and in the table are reported 9.

The number of local recurrences is nine, we have changed the number in the legend

Reviewer 2 Report

Manuscript ID: cancers-2209034 - Review Report

Thank you for this very interesting and relevant study.  As an observational study it addresses a very relevant subject for further investigations. However, as a study claiming to change future management of chondrosarcomas (line 248-249) the study design and results are too limited, and some issues needs to be addressed as well as the conclusion needs to be modified.

The statement and interpretation if grade affect DSS is confusing.

Title

The last part does not address the research question. Also, only 1 patient with grade 2 had amputation.

Abstract

Line 45: please see line 160

Introduction

Line 85-87: Taken the stated conclusion into consideration a clarification of aims or hypotheses is needed.

Line 87: the whole skeleton or the appendicular skeleton?

Material and Methods

Overall, this is not a population-based cohort. It is a small sample size relative to number of Centres involved and inclusion time-period. The inclusion criteria are narrow (line 235-236) and the cohort is not consecutive according to mentioned inclusion criteria (line 235-236). The selection bias needs to be addressed in discussion/limitations.

Furthermore, were the treatment regimes the same at the three Centres throughout the whole inclusion time-period.

Eg.  a clear statement of all inclusion criteria and exclusion criteria or even a flowchart of the inclusion process would clarify

Definition of outcome measures is needed since the terms are often confused and misinterpreted. Also how was LR defined/diagnosed? Was the definition and diagnostic of LR the same at all Centres in the whole inclusion time-period?  

Results

Table 1                                                                                                                                                                  Why did 54% of patients with Grade 1 have wide resection? Was that a choice for large tumors? And an explanation the low LR rate in Grade 1? Please address this in the discussion

Line 148: In the table 9 patients are reported to have LR

Line 157-158: please comment on SHR=16.3% relative to CI:2.7-97.9. What is the impact of a p value and SHR with this wide confidence-interval. Is the cohort representative for this comparison?

Line 160: see above comment

Line 165: why “disease specific failure”?

Discussion

Line 181-182: This is a confusing statement since you demonstrated the opposite in line 160

Line 196: same as above.

Line 196-197: could this be explained by selection bias? Were some patients with CS grade 2 referred to the oncologist at time of diagnosis and not offered surgery? Or were some patients not offered surgery for other causes.

Line 208: You demonstrate that grade 2 CS is locally more aggressive with a higher rate of LR and combine this with the low rate of DSS in grade 2 CS. Please comment on this statement relative to the findings in line 160.

Line 209-210: However, when you compare grades by competing risk analysis, you find the opposite: SHR=16.3 (line 157 and 227-228)

Line 221: same as above

Conclusion

Line 246-247: same as above. When you do the competing risk analysis grade is statistically significant for poor DSS (line 157 and 227-228)

Again, this not a population-based cohort. It is a retrospective study with a small not consecutive sample size exposed to selection bias. The design and results are not appropriate for a change in treatment management of CS. Thus, the conclusion needs modification into a not causal statement.

Author Response

Reviewer 2

Thank you for this very interesting and relevant study.  As an observational study it addresses a very relevant subject for further investigations. However, as a study claiming to change future management of chondrosarcomas (line 248-249) the study design and results are too limited, and some issues needs to be addressed as well as the conclusion needs to be modified.

The statement and interpretation if grade affect DSS is confusing.

Title

The last part does not address the research question. Also, only 1 patient with grade 2 had amputation.

We have changed the title to describe more accurately our results.

Abstract

Line 45: please see line 160

The text has been changed as: LR in grade 2 CS does not.

Introduction

Line 85-87: Taken the stated conclusion into consideration a clarification of aims or hypotheses is needed.

The purpose of this study has been rephrased.

Line 87: the whole skeleton or the appendicular skeleton?

The purpose of this study has been rephrased and this sentence as belonging more to the discussion has been removed.

Material and Methods

Overall, this is not a population-based cohort. It is a small sample size relative to number of Centres involved and inclusion time-period. The inclusion criteria are narrow (line 235-236) and the cohort is not consecutive according to mentioned inclusion criteria (line 235-236). The selection bias needs to be addressed in discussion/limitations.

Missing images and thereby excluded patients were mainly timed to years before 2020. Since 2020 all consecutive patients were included in this study thereby decreasing the selection bias. This has been addressed more carefully in the limitations.

Furthermore, were the treatment regimes the same at the three Centres throughout the whole inclusion time-period.

The three centres share very similar treatments strategies. The regimens did not significantly change between centres and over time.

Eg.  a clear statement of all inclusion criteria and exclusion criteria or even a flowchart of the inclusion process would clarify

A flowchart has been added.

Definition of outcome measures is needed since the terms are often confused and misinterpreted. Also how was LR defined/diagnosed? Was the definition and diagnostic of LR the same at all Centres in the whole inclusion time-period? 

A definition of primary and secondary outcome measures have been written more clearly and definition of LR has been rewritten for clarity.

Results

Table 1                                                                                                                                                                  Why did 54% of patients with Grade 1 have wide resection? Was that a choice for large tumors? And an explanation the low LR rate in Grade 1? Please address this in the discussion

Line 148: In the table 9 patients are reported to have LR

Nine patient had LR, we have corrected this mistake.

Line 157-158: please comment on SHR=16.3% relative to CI:2.7-97.9. What is the impact of a p value and SHR with this wide confidence-interval. Is the cohort representative for this comparison?

Thank you for this comment. Using p-values has a long history though we know the problem with the p-value is not the p-value itself. The problem is ignorance about statistical inference, i.e., about the principles for using empirical observations and statistical reasoning to arrive at scientifically sound conclusions.

Confidence intervals represent a superior way to present generalization uncertainty. Confidence intervals have the advantage of measuring the uncertainty of the size of an estimated effect.  The wide confidence interval demonstrates these problems and are therefore presented so that the reader can estimate the value of results him/herself. Reporting  p-value with confidence intervals is the right way to present results and we do feel that this cohort is as representative for this comparison as possible. Our problem with many other writers is to use word statistical significance when p-value is merely developed for performing rational generalization of findings, not for describing data as we should discuss about uncertainty. However, at the moment we are caught in the history and present results with p-values, confidence interval.

Line 160: see above comment

Line 165: why “disease specific failure”?

Competing risk analysis measures failures not survival.

Discussion

Line 181-182: This is a confusing statement since you demonstrated the opposite in line 160

Line 196: same as above.

We thank the reviewer for this important comment. Indeed the text is confusing, since one important results was missing. When stratified by grade the role of LR in DSS was not significant. We have added this result and changed the discussion to clarify the text accordingly.

Line 196-197: could this be explained by selection bias? Were some patients with CS grade 2 referred to the oncologist at time of diagnosis and not offered surgery? Or were some patients not offered surgery for other causes.

We thank the reviewer for this comment. We have discussed this in our study group extensively and we do not have a clear explanation for this. All centres share a very surgical approach to chondrosarcoma, and none of the patients is referred to oncologist unless the disease has largely metastasized. Only one patient refused from surgery and surgery was offered and accepted by all the other patients.

Line 208: You demonstrate that grade 2 CS is locally more aggressive with a higher rate of LR and combine this with the low rate of DSS in grade 2 CS. Please comment on this statement relative to the findings in line 160.

Please see above.

Line 209-210: However, when you compare grades by competing risk analysis, you find the opposite: SHR=16.3 (line 157 and 227-228)

Please see above.

Line 221: same as above

Please see above.

Conclusion

Line 246-247: same as above. When you do the competing risk analysis grade is statistically significant for poor DSS (line 157 and 227-228)

Again, this not a population-based cohort. It is a retrospective study with a small not consecutive sample size exposed to selection bias. The design and results are not appropriate for a change in treatment management of CS. Thus, the conclusion needs modification into a not causal statement.

We agree with the reviewer that this is not a population-based cohort. However, in sarcoma surgery, given the rarity of the tumour and the huge variation in diagnosis, treatment algorithms and outcome measures, to undertake a population based study would require years to answer any meaningful questions. We are therefore, left with retrospective studies from institutions with a similar approach to management. This is reflected in the small sample size included in this study, which is however, the largest series for this tumour location.  As far as possible, we have tried to exclude any selection bias by using inclusion and exclusion criteria common to all centres. However, we do acknowledge that despite our best efforts, some bias on the part of individual centres may exist.

Reviewer 3 Report

I congratulate the authors on their work. I enjoyed reading the manuscript as it really touches my own deliberations in my consultation and many questions still remain unanswered. This is a quite large collective given the really rare condition of the disease. I thus would like this article to be published. With respect to data presentation and conclusions drawn from the data I still have some concerns and I would encourage the authors to still improve on that point.

Abstract: The combination of mean and range sound at first glance a bit unusual. The abstract is text without a formal structure. Therefore an adjunct might help to signal, that the results part transitions to the conclusion part.

Lines 72-75. Hard to follow, please rephrase.

Study hypothesis: You did not evaluate the behaviour elsewhere so you cannot compare directly. You evaluated it in the proximal humerus to clarify how it behaves in that specific location. In the discussion you can then compare it with the literature. This is, however, something different. Study hypothesis needs to be clear.

I am not sure what gain of information is obtained by a prospectively maintained database. The data are retrospective. That is all there is to say.

Statistical analysis: You state you report continuous variables as mean and 95%CI, yet you report age for example as mean and range. Was age normally distributed? How was normality verified? Same applies for follow-up.

Since there are 5 cases with an intralesional resection on a G2 tumour, the whole discussion on treatment necessities with respect to LR are based on these five cases. Maybe it would be worth to describe these in more detail with respect to when and if LR occurred and how long follow up was in these patients. Of note: reading the data presented in your second table thoroughly, only one local recurrence was after intralesional surgery. Given that 5 intralesional curettages were performed in grade 2 tumours and those seemed to be LR free I fail to follow the conclusions drawn by the authors.

In M&M the authors state that the minimum follow-up was 2 years to be included in the study. In the results section it becomes clear that evidently at least three patients were included with a follow-up much shorter than 2 years (4, 7, and 11 months). The data presentation thus needs to be corrected.

The Kaplan-Meier curve conveys a strongly biased picture, since drop-outs are not reported and as such the curve presented is a best-case scenario. It would only be fair to at leas provide a worst-case scenario curve as supplementary data. It also is not clear why the display of the two groups differs so much with respect to follow-up. This would somehow imply, that all the grade 3 patients were so bedstruck they could not attend their appointments any more. Please be more thorough with respect to the Kaplan-Meier presentation.

Is local recurrence a predictor for or associated with metastasis formation?

Lines 174 and 175. I am not sure you demonstrated this since it was not comparative study. Your study collective showed less aggressive courses maybe when compared to other locations reported in the literature. This is a bit hairsplitting, scientifically this is, however, a big difference. Since the topic is so crucial, I suggest to stay clearly objective.

Lines 183-186. Here would be a good opportunity to report findings of other locations from other studies so as to back up your statement that a location at the proximal humerus is less problematic.

Lines 187. In our study collective.... no patient graded 1 died

The necessary reason for intralesional curettage would be that even with this technique the rate of LR is low. In grade 2 they also do not die and rarely develop metastasis and yet curettage seems not to be accepted.

Lines 198 ss: margins not significant. Given the low subgroup sample size a statistically significant effect can not necessarily be expected. Since the topic is quite delicate, I suggest to focus also on the descriptive statistics since readers not so versed in statistics might misinterpret such a statement. Just because you did not see a statistical significance does not mean that there is no difference since - due to the rarity of the condition - your study is underpowered. Also descriptive statistics can be very powerful when correctly presented. Altough this is just my personal opinion, I find the notion that margins might be irrelevant just because this aspect did not reach statistical significance in your data set a bit perturbing. You also mention all this correctly in your study limitations. I just encourage you to not lose the focus on the descriptive statistics.

Line 205: Very good point.

What do the authors mean by: "radiological and pathological data were comprehensive". Usually the grading is a histopathological feature. I guess I know what the authors mean but maybe this could be specified.

Line 243: Since the authors never mention how it behaves at other locations it remains unclear how this conclusion was drawn.

Line 249: From my reading the data I am not sure this statement is backed up. Please double-check.

Author Response

Reviewer 3

I congratulate the authors on their work. I enjoyed reading the manuscript as it really touches my own deliberations in my consultation and many questions still remain unanswered. This is a quite large collective given the really rare condition of the disease. I thus would like this article to be published. With respect to data presentation and conclusions drawn from the data I still have some concerns and I would encourage the authors to still improve on that point.

Abstract: The combination of mean and range sound at first glance a bit unusual. The abstract is text without a formal structure. Therefore an adjunct might help to signal, that the results part transitions to the conclusion part.

Thank you for highlighting this. We have included means and ranges as the preference of the journal based on the required statistical outcomes. The abstract has been paragraphed to make for easier reading.

Lines 72-75. Hard to follow, please rephrase.

The text has changed for clarity

Study hypothesis: You did not evaluate the behaviour elsewhere so you cannot compare directly. You evaluated it in the proximal humerus to clarify how it behaves in that specific location. In the discussion you can then compare it with the literature. This is, however, something different. Study hypothesis needs to be clear.

Noted by other reviewers as well, the purpose of the study has been changed.

I am not sure what gain of information is obtained by a prospectively maintained database. The data are retrospective. That is all there is to say.

We agree with reviewers comment and we have changed the text accordingly. The purpose of this name prospectively maintained database is to describe that our database is consecutive.

Statistical analysis: You state you report continuous variables as mean and 95%CI, yet you report age for example as mean and range. Absolute correct-we have changed the text accordingly. Was age normally distributed?. How was normality verified? Same applies for follow-up.

Age was normally distributed. Normality was tested using Shapiro-Wilkins test. This has been added to the text.

The age distribution of follow up was influenced by the age of presentation and as such was not a normality variable other than to state that the age of presentation, which was normally distributed, influenced the age of follow up. The most important assumption of Cox regression model is the proportional hazard ratio, distribution of follow-up time has no importance.

Since there are 5 cases with an intralesional resection on a G2 tumour, the whole discussion on treatment necessities with respect to LR are based on these five cases. Maybe it would be worth to describe these in more detail with respect to when and if LR occurred and how long follow up was in these patients.

We have added more details about these patients in the text and the appearance of LR in grade 2, as well as in all grades is shown in figure 2 (previously figure 1).

Of note: reading the data presented in your second table thoroughly, only one local recurrence was after intralesional surgery. Given that 5 intralesional curettages were performed in grade 2 tumours and those seemed to be LR free I fail to follow the conclusions drawn by the authors.

We have deleted this conclusion.

In M&M the authors state that the minimum follow-up was 2 years to be included in the study. In the results section it becomes clear that evidently at least three patients were included with a follow-up much shorter than 2 years (4, 7, and 11 months). The data presentation thus needs to be corrected.

Thank you for highlighting this issue. The minimum 2 year follow up period applied to those alive for the purposes of the survival analysis.

The Kaplan-Meier curve conveys a strongly biased picture, since drop-outs are not reported and as such the curve presented is a best-case scenario. It would only be fair to at leas provide a worst-case scenario curve as supplementary data. It also is not clear why the display of the two groups differs so much with respect to follow-up. This would somehow imply, that all the grade 3 patients were so bedstruck they could not attend their appointments any more. Please be more thorough with respect to the Kaplan-Meier presentation.

We have changed the figure to a new KM figure presenting the numbers at risk for each grade. In our opinion this visualizes more clearly the differences in curves.

Is local recurrence a predictor for or associated with metastasis formation?

In this material we could not show any association of between local recurrence and metastasis. Indeed, we have shown that even where local recurrence does not occur, patients with higher grade chondrosarcoma can succumb to their disease. Whilst the inverse does not prove the former, the relationship between LR and overall survival for higher grade chondrosarcoma appears to be independent on the basis of our data.

Lines 174 and 175. I am not sure you demonstrated this since it was not comparative study. Your study collective showed less aggressive courses maybe when compared to other locations reported in the literature. This is a bit hairsplitting, scientifically this is, however, a big difference. Since the topic is so crucial, I suggest to stay clearly objective.

We have rephrased the sentence for clarity.

Lines 183-186. Here would be a good opportunity to report findings of other locations from other studies so as to back up your statement that a location at the proximal humerus is less problematic.

We thank the reviewer for this suggestion and have added text accordingly.

Lines 187. In our study collective.... no patient graded 1 died

The necessary reason for intralesional curettage would be that even with this technique the rate of LR is low. In grade 2 they also do not die and rarely develop metastasis and yet curettage seems not to be accepted.

Text has been changed for clarity

Lines 198 ss: margins not significant. Given the low subgroup sample size a statistically significant effect can not necessarily be expected. Since the topic is quite delicate, I suggest to focus also on the descriptive statistics since readers not so versed in statistics might misinterpret such a statement. Just because you did not see a statistical significance does not mean that there is no difference since - due to the rarity of the condition - your study is underpowered. Also descriptive statistics can be very powerful when correctly presented. Altough this is just my personal opinion, I find the notion that margins might be irrelevant just because this aspect did not reach statistical significance in your data set a bit perturbing. You also mention all this correctly in your study limitations. I just encourage you to not lose the focus on the descriptive statistics.

We do agree with the reviewer and we have changed the text accordingly.

Line 205: Very good point.

What do the authors mean by: "radiological and pathological data were comprehensive". Usually the grading is a histopathological feature. I guess I know what the authors mean but maybe this could be specified.

We have changed the text for clarity

Line 243: Since the authors never mention how it behaves at other locations it remains unclear how this conclusion was drawn.

We have changed the text for clarity

Line 249: From my reading the data I am not sure this statement is backed up. Please double-check.

We have changed the text for clarity

Reviewer 4 Report

This study is intriguing research that compiles a considerable number of cases limited to chondrosarcoma originating near the proximal humerus. However, it appears that data is insufficient to discuss treatment approaches based on the grade of the sarcoma. The following improvements would lead to a more favorable report.

1.       This paper contains various errors, including grammatical ones. A comprehensive editing process is imperative to ensure its accuracy.

2.       On line 64, the authors indicate that the incidence of ACT has undergone a more than tenfold increase. However, the referenced article #7 does not appear to reflect this substantial rise in incidence.

3.       It is recommended that the authors mention the incidence of transformation into a higher grade in addition to LR.

4.       The most important aspect of this study, which is limited to tumors of the upper arm bones, is that tumors in the proximal humerus are expected to have different prognoses and behaviors compared to those in other regions, thereby warranting different therapeutic approaches. However, no evidence is provided in the current study to support this notion, and no such information can be gleaned from reference 8, which would serve as the basis for such a conclusion. Moreover, this literature cited as the author's prior research is not authored by any of the current authors or institutions involved in this study.

5.       The title is intriguing and attention-grabbing, however, the data on amputation for grade 2 chondrosarcoma appears too limited to draw a definitive conclusion. It is speculated that cases, where amputation is selected, are in highly unique circumstances, however, such a detailed clinical context is difficult to deduce from this study.   

6.       Please reconsider the title of Table 2.

7.       In Line 181, the authors assert that the presence of LR was not a statistically significant prognostic factor for decreased DSS. This conclusion appears to contradict the results reported in Line 160. Furthermore, LR was observed to be significantly correlated with mortality (Table 2).

8.       Is the statement in Line 194-195 drawn from this study or from other reports cited?

9.       The intention and basis behind lines 198 to 202 remain unclear.

Author Response

Reviewer 4

This study is intriguing research that compiles a considerable number of cases limited to chondrosarcoma originating near the proximal humerus. However, it appears that data is insufficient to discuss treatment approaches based on the grade of the sarcoma. The following improvements would lead to a more favorable report.

  1. This paper contains various errors, including grammatical ones. A comprehensive editing process is imperative to ensure its accuracy.

The text has been grammatically checked and corrected accordingly.

  1. On line 64, the authors indicate that the incidence of ACT has undergone a more than tenfold increase. However, the referenced article #7 does not appear to reflect this substantial rise in incidence.

In the article from van Praag et al the incidence has been less than 1.0 in 90’s and in 2012 the incidence is very close to 10 per million. Since the decimal are not presented, we have changed the text accordingly.

  1. It is recommended that the authors mention the incidence of transformation into a higher grade in addition to LR.

The grade of the tumour was always the same as in the primary tumour. This has been added to the text.

  1. The most important aspect of this study, which is limited to tumors of the upper arm bones, is that tumors in the proximal humerus are expected to have different prognoses and behaviors compared to those in other regions, thereby warranting different therapeutic approaches. However, no evidence is provided in the current study to support this notion, and no such information can be gleaned from reference 8, which would serve as the basis for such a conclusion. Moreover, this literature cited as the author's prior research is not authored by any of the current authors or institutions involved in this study.

We are sorry that this point has not been made clear. We have amended the manuscript to hopefully clarify this position. It is the notion of this study that when compared to other tumour locations, CS of the proximal humerus seem to behave in a less aggressive manner, at least for grade 1 and 2 tumours where a low incidence of LR is seen, even with intralesional or marginal resections. The development of LR is comparatively low when compared to other anatomical locations and does not appear to affect DSS. Therefore, if the margin appears less instrumental at the development of LR, and the development of LR does not affect DSS, we would suggest that the treatment approach can instead have more of a bearing on preserving function rather than trying to achieve wider margins, sacrificing function but not affecting the LRFS nor DSS.

  1. The title is intriguing and attention-grabbing, however, the data on amputation for grade 2 chondrosarcoma appears too limited to draw a definitive conclusion. It is speculated that cases, where amputation is selected, are in highly unique circumstances, however, such a detailed clinical context is difficult to deduce from this study.

We agree the title was overselling and we have changed it

  1. Please reconsider the title of Table 2.

Changed for clarity

  1. In Line 181, the authors assert that the presence of LR was not a statistically significant prognostic factor for decreased DSS. This conclusion appears to contradict the results reported in Line 160. Furthermore, LR was observed to be significantly correlated with mortality (Table 2).

We thank the reviewer for this important comment. Indeed the text is confusing, since one important results was missing. When stratified by grade the role of LR in DSS was not significant. We have added this result and changed the discussion to clarify the text accordingly.

  1. Is the statement in Line 194-195 drawn from this study or from other reports cited?

We thank for the reviewer’s accuracy. This statement was from previous versions and was deleted.

  1. The intention and basis behind lines 198 to 202 remain unclear.

The results for this statement are presented in lines 145-146. Lines which are hidden between Figure 1 (Now Figure 2) and table 2.

Round 2

Reviewer 3 Report

Revision 2.

Although the authors did not change much in their manuscript, it has still strongly improved, since critical statements were adjusted. The presentation of the Kaplan-Meier curves still seems unacceptable to me though. This should still be addressed by the authors. The other comments from my part are just minor aspects.

Abstract: Disease specific survival was affected by grade (p=0).

p=0 is statistically almost impossible. maybe <0.001

Otherwise good abstract

Introduction: no further comments

M&M: I guess it is Shapiro-Wilk test, not Wilkins....

I also still have a hard time getting used to the combination of mean and range, but if that is a requirement of the journal I suppose it is up to the journal to decide why they want such an unusual presentation.

Results:

With respect to the Kaplan-Meier curve, I keep up my statement that I completely miss a transparent display of cases lost to follow-up. I acknowledge that the chances a tumour patient just goes to a different hospital and thus is lost in the data set are much lower than in conventional orthopaedic conditions such as periprosthetic joint infection. Yet, the present form of displaying the Kaplan-Meier curve might be highly misleading. Whether it actually is misleading or not, I cannot judge without the original data. It is up to the authors to convice the readers that it is not. This can be done in a first step by highlighting the dropouts. If there are many dropouts, a second worst-case scenario curve has to be presented. The way how the curve reads itself right now is, that for example in figure 4, follow-up of all CS2 patients was for over 25 years. I find this unlikely.

Line 164. Isn't it Pearson?

Discussion:

We have shown... that means you sort of proved it. As already stated in my previous review, you cannot prove anything when comparing with other sites since you present no data of these other sites. In your study collective, you observed a local recurrence rate of XYZ and a DSS of ZYX. In the literature, other studies described ABC and CBA. Comparing these values it would seem that CS at the proximal humerus can be considered as more benign.....

Comparison with the pelvis is fair. Yet, it has been traditionally known that a location at the pelvis has the worst prognosis. It would thus be much more interesting to compare the proximal humerus with the proximal femur.

Lines 216. I find this an interesting and noteworthy statement.

Lines 219-233. Very interesting discussion.

Line 245-247. I still find it hard to logically follow these sentences. I offer you here a wording suggestion. Would this be correct? "In our descriptive analyses, we saw a lower rate of LR when margins were improved. This effect failed, however, to reach statistical difference, possibly due to the low sample size in the subgroups."

Lines 263-265. Very interesting.

Author Response

Thank you for taking the time to review our manuscript. We are grateful for the feedback and welcome the opportunity to amend and clarify the issues raised by the reviewers. We have addressed in detail below, the concerns raised by the reviewers and we hope this is satisfactory.

Although the authors did not change much in their manuscript, it has still strongly improved, since critical statements were adjusted. The presentation of the Kaplan-Meier curves still seems unacceptable to me though. This should still be addressed by the authors. The other comments from my part are just minor aspects.

Abstract: Disease specific survival was affected by grade (p=0).

p=0 is statistically almost impossible. maybe <0.001

Otherwise good abstract

Response: This has been changed accordingly.

Introduction: no further comments

M&M: I guess it is Shapiro-Wilk test, not Wilkins....

Response: This has been changed accordingly.

I also still have a hard time getting used to the combination of mean and range, but if that is a requirement of the journal I suppose it is up to the journal to decide why they want such an unusual presentation.

Results:

With respect to the Kaplan-Meier curve, I keep up my statement that I completely miss a transparent display of cases lost to follow-up. I acknowledge that the chances a tumour patient just goes to a different hospital and thus is lost in the data set are much lower than in conventional orthopaedic conditions such as periprosthetic joint infection. Yet, the present form of displaying the Kaplan-Meier curve might be highly misleading. Whether it actually is misleading or not, I cannot judge without the original data. It is up to the authors to convice the readers that it is not. This can be done in a first step by highlighting the dropouts. If there are many dropouts, a second worst-case scenario curve has to be presented. The way how the curve reads itself right now is, that for example in figure 4, follow-up of all CS2 patients was for over 25 years. I find this unlikely.

Response: We do apologise that we have not been able to clarify this result. We hope now to have rectified this situation. To clarify, all patients had continuous follow up until the point of last clinical assessment or to the time of death. In 2 cases, the point of final follow up or death was missing. This has now been amended and the details of these patients added. Having rectified these issues, we remain of the opinion that the Kaplan-Meier method remains the best method for presenting these results. We have added to the figures, the number of patients at risk which we believe now allows the reader to understand better the number of patients remaining for analysis at each timepoint. We hope this now clarifies this issue sufficiently.

Line 164. Isn't it Pearson?

Response: This has now been changed accordingly.

Discussion:

We have shown... that means you sort of proved it. As already stated in my previous review, you cannot prove anything when comparing with other sites since you present no data of these other sites. In your study collective, you observed a local recurrence rate of XYZ and a DSS of ZYX. In the literature, other studies described ABC and CBA. Comparing these values it would seem that CS at the proximal humerus can be considered as more benign.....

Response: The results presented demonstrate the findings as discussed. The reviewer is correct that we have not proven anything, but we have only demonstrated our results. To prove whether we are accurate in our results, the findings would need to be ratified by other institutions. As discussed, the number of patients, given the rarity of the disease, could not be demonstrated to any degree of significance if we had used patients and data from any one institution. It is the fact that we have pooled data from 3 centres with comparable methods of data collection and treatment algorithms, that allows to demonstrate the findings with a degree of significance. We have positioned our findings against the available literature but as can be seen, the numbers included in our study exceed those of other studies, and in fact we have looked at variables not included in other studies. Therefore, the reviewer is correct that the results we present are novel and therefore not yet ratified by other, independent centres.

Comparison with the pelvis is fair. Yet, it has been traditionally known that a location at the pelvis has the worst prognosis. It would thus be much more interesting to compare the proximal humerus with the proximal femur.

Response: Thank you for this comment with which we fully agree. Indeed, we tried to find other locations other than pelvis. However, we were unable to identify studies that looked solely at the proximal femur or any other specific tumour location. In fact, all the studies identified included the proximal femur or indeed the proximal humerus, as part of a disease specific article (for example, looking at chondrosarcoma as a whole, or looking at cartilaginous tumours of bone as a group). Even when looking at review articles, where it may have been possible to extract specific tumour locations, the numbers remained very low and therefore, comparison was not possible.  

Lines 216. I find this an interesting and noteworthy statement.

Lines 219-233. Very interesting discussion.

Line 245-247. I still find it hard to logically follow these sentences. I offer you here a wording suggestion. Would this be correct? "In our descriptive analyses, we saw a lower rate of LR when margins were improved. This effect failed, however, to reach statistical difference, possibly due to the low sample size in the subgroups."

Response: We have re-read our sentence and compared it to the sentence the reviewer suggested. We still feel that our sentence reflects our opinions  better. However, if our sentence is not acceptable, we change it accordingly.

Lines 263-265. Very interesting.

Reviewer 4 Report

The authors have not provided concrete data to substantiate their assertion that chondrosarcomas situated in the proximal humerus are unique in their slow progression and necessitate individual analysis. Comparing chondrosarcomas in the proximal humerus with those in other appendicular regions, such as the proximal femur, using institution-specific data would have been more substantial. And without these results, this study would not be meaningful. If the conclusion that limb-sparing surgery is preferable for proximal humerus chondrosarcoma is true, it would be very interesting. However, I believe that the number of events for G2 chondrosarcoma is still too small to draw such a conclusion.

The authors should examine the association between marginal status and survival (if they retain this title). They should also clarify how they define "marginal" and "wide" margins. The marginal margin between the tumor and soft tissue does not appear to have as much effect on local recurrence because lobules of the chondroid matrix are frequently enclosed by thin fibrous septa. The marginal margin between tumors and bone marrow, on the other hand, appears to be more problematic, although I personally find it difficult to marginally resect central CS from bone marrow with the intent.

Is the reported p-value of 0 accurate (line 44)? Does it mean p<0.0001 and so on?

Please explain the methodology and results of the stratified analysis (line 175). It is unsurprising that subgroup analysis would lose statistical significance due to the smaller sample size.

Did any of the intralesional/marginal CS 2 and 3 undergo additional resection?

Please kindly provide the references upon which the statement made in lines 219-226 is predicated.

Author Response

Thank you for taking the time to review our manuscript. We are grateful for the feedback and welcome the opportunity to amend and clarify the issues raised by the reviewers. We have addressed in detail below, the concerns raised by the reviewers and we hope this is satisfactory.

Reviewer 4

The authors have not provided concrete data to substantiate their assertion that chondrosarcomas situated in the proximal humerus are unique in their slow progression and necessitate individual analysis. Comparing chondrosarcomas in the proximal humerus with those in other appendicular regions, such as the proximal femur, using institution-specific data would have been more substantial. And without these results, this study would not be meaningful. If the conclusion that limb-sparing surgery is preferable for proximal humerus chondrosarcoma is true, it would be very interesting. However, I believe that the number of events for G2 chondrosarcoma is still too small to draw such a conclusion.

Response: We believe it is important to study anatomic localizations as their own entity. If we were discussing the pelvis and stating that their behaviour was different to other locations, that would not be contentious. However, to assume that all anatomical locations behave in the same way is effectively overlooking the potential for patient specific treatment algorithms. How many patients would be overtreated if we were to continue with this assumption? We acknowledge the reviewer’s opinion that individual institution data would be more powerful than pooled data, but by the reviewer’s own admission, this would result in small numbers making any findings meaningless. Should we simply ignore our findings and assume that all locations, outside the pelvis behave in the same way? We do not believe that is correct and we have presented results which support our hypothesis.

The authors should examine the association between marginal status and survival (if they retain this title). They should also clarify how they define "marginal" and "wide" margins. The marginal margin between the tumor and soft tissue does not appear to have as much effect on local recurrence because lobules of the chondroid matrix are frequently enclosed by thin fibrous septa. The marginal margin between tumors and bone marrow, on the other hand, appears to be more problematic, although I personally find it difficult to marginally resect central CS from bone marrow with the intent.

Response: The margin was quantified by specialist bone sarcoma pathologist and classified according to the system described by Enneking in 1990. This has been added to the text. We agree with the reviewer that the definition of margin is extremely difficult as the width of a wide margin or marginal margin has never been accurately defined and discrepancy exists in the practical definition of a wide or marginal margin. However, the Enneking definition is the most widely used and therefore to maintain continuity with the available literature, we have continued this definition.

The role of marginal status and survival has been added to the text with an additional figure.

Is the reported p-value of 0 accurate (line 44)? Does it mean p<0.0001 and so on?

Response: This has been changed accordingly.

Please explain the methodology and results of the stratified analysis (line 175). It is unsurprising that subgroup analysis would lose statistical significance due to the smaller sample size.

Response: Additional information of competing risk analysis has been added to the material and methods section for clarity.

Did any of the intralesional/marginal CS 2 and 3 undergo additional resection?

Response: No. One patient with a grade 3 CS treated with intralesional surgery declined further surgery due to her age. She died 6 months later due to unrelated health issues. All patients with grade 2 tumours treated with intralesional margins were followed without additional surgical treatment.

Please kindly provide the references upon which the statement made in lines 219-226 is predicated.

Response: This has been added accordingly.